# The Microbiota of the Outer Gut Mucus Layer of the Migrating Northeast Arctic Cod (*Gadus morhua*) as Determined by Shotgun DNA Sequencing

**DOI:** 10.3390/microorganisms12112204

**Published:** 2024-10-31

**Authors:** Typhaine Le Doujet, Peik Haugen

**Affiliations:** Department of Chemistry and the Center for Bioinformatics, Faculty of Science and Technology, UiT The Arctic University of Norway, N-9037 Tromsø, Norway; typhaine.l.doujet@uit.no

**Keywords:** Northeast Arctic cod microbiome, adherent bacteria, autochthonous, mucus microbiota, intestinal tract, NGS, metagenome assembled genomes, MAGs

## Abstract

Animals form functional units with their microbial communities, termed metaorganisms. Despite extensive research on some model animals, microbial diversity in many species remains unexplored. Here, we describe the taxonomic profile of the microbes from the outer gut mucus layer from the Northeast Arctic cod using a shotgun DNA sequencing approach. We focused on the mucus to determine if its microbial composition differs from that of the fecal microbiota, which could reveal unique microbial interactions and functions. Metagenomes from six individuals were analyzed, revealing three different taxonomic profiles: Type I is dominated in numbers by *Pseudomonadaceae* (44%) and *Xanthomonadaceae* (13%), Type II by *Vibrionaceae* (65%), and Type III by *Enterobacteriaceae* (76%). This stands in sharp contrast to the bacterial diversity of the transient gut content (i.e., feces). Additionally, binning of assembled reads followed by phylogenomic analyses place a high-completeness bin of Type I within the *Pseudomonas fluorescens* group, Type II within the *Photobacterium phosphoreum* clade, and Type III within the *Escherichia/Shigella* group. In conclusion, we describe the adherent bacterial diversity in the Northeast Arctic cod’s intestine using shotgun sequencing, revealing different taxonomic profiles compared to the more homogenous transient microbiota. This suggests that the intestine contains two separate and distinct microbial populations.

## 1. Introduction

Animals live in close dynamic relationships with communities of microorganisms to form what have been named metaorganisms (or holobionts) [1,2]. In humans, the microbiota with the highest density of microorganisms is found in the intestinal tract [1,3,4], where it plays essential roles in food digestion, host immunity, host metabolism, and stress responses [1,5]. The structure of the gut microbial community can depend on various factors such as diet, habitats, host lineages, and external stimuli. In vertebrates, phyla such as Firmicutes, Bacteroidetes, Actinobacteria, Proteobacteria (newly proposed to be renamed to “Bacillota”, “Bacterioidota”, “Actinoycetota”, and “Pseudomonadota”, respectively, by the International Committee on Systematics of Prokaryotes [6]) and Fusobacteria generally dominate, but with variations of their relative proportions [7,8]. For example, the gut of most mammals is typically rich in representatives from Bacteroidetes and Firmicutes [7,9,10], whereas other types of bacteria densely populate the gut of reptiles, fish, and birds [11]. In fish, the gut hosts predominantly Proteobacteria [3,8,12,13].

An interesting dimension of the gastrointestinal tract, of at least some model mammals, is that the intestinal lumen and mucus layer host separate microbiota that are considered to be autochthonous (i.e., adherent bacteria) and allochthonous (i.e, non-adherent bacteria) [14]. Whereas the allochthonous bacteria are in direct contact with the digesta within the intestinal lumen, the autochthonous bacteria populate the outer mucus layer that covers the inner mucus layer and the intestinal epithelium. The reason why these two microbiota are separate is due to the complex structure and function of the intestine [15]. The focus herein will remain mainly on the adherent microbial community that populates the mucin-rich mucus. Mucin is the main component of the intestinal mucus in animals and is mainly made of O-glycosylated proteins [14,16]. Such proteins can serve as a source of energy for microorganisms with genes encoding catabolic glycosylic enzymes. Thus, bacteria capable of utilizing mucin as a carbon source may outcompete those that lack this capability [14].

Microorganisms that inhabit the mucus are critical to the health of the intestine. They are in close proximity to the host epithelium, and here they work in symbiosis with the host, ideally without triggering the immune system. Firstly, they form a protective barrier against pathogens [17,18]. Secondly, they also serve critical roles in the interplay with the host, e.g., in digestion, immunity, and nutrient exchange [5]. Today, it is clear that not only the physiology of the intestine is affected by the gut microbiota. The microbial composition also affects the whole-body metabolism by communicating with distant organs like the brain, liver, and heart [14,15]. This is made possible due to bidirectional exchanges of small molecules, including those produced by microbes, between the outer mucus layer and the host epithelium via goblet cells [14]. Once molecules penetrate the epithelium barrier, they can enter the bloodstream and make their way to all organs.

In comparison with humans, less is known about the gut microbiota of fish, particularly regarding the microbial structure and their roles in intestinal health. One of the best-studied fish models in this respect is zebrafish (*Danio rerio*). This is because the zebrafish offers several benefits to study the effect of the gut microbiota on the health of the host, and this has led to increased attention to how such in vivo models can contribute to a wider understanding in this field [19].

In contrast to studies that involve advanced fish models such as Zebrafish, studies on the gut microbiota of economically important reared or wild-caught fish must typically rely on classical microbiological methods or DNA sequencing methods. The latter can be done by purifying DNA from the gut and then amplifying parts of the 16S rDNA (amplicon sequencing), or by sequencing the total DNA directly (shotgun sequencing; metagenomics). To succeed with the latter, sufficient high-quality total DNA is extracted from feces or mucus material, which is a challenging task and therefore not typically done. We and others have during recent years used DNA sequencing methods to study the microbial composition of the gut of different populations of Atlantic cod (e.g., [20,21,22]).

However, these studies are mainly focusing on the non-adherent microorganisms (allochthonous) of the gastrointestinal tract with the conclusion that *Photobacterium* is the most abundant bacterium. It is however still unclear how the adherent bacteria vary in composition compared with the non-adherent bacteria.

The Northeast Arctic cod (NEAC) is one of the world’s largest cod stocks and has substantial economic value, supporting numerous commercial fisheries across the Atlantic, which are vital for the economies of coastal communities [23]. Thus, being a key species in marine ecosystems and fisheries. It plays a crucial role in the trophic dynamics of the marine food web, serving as both predator and prey [24]. This species significantly influences the population structures of other marine organisms and helps maintain the balance of the ecosystem. Additionally, NEAC is of historical and cultural significance, which further underscores its importance, making its management and conservation a priority for sustainable marine resource utilization [25].

Here, we have established the first bacterial profiles of autochthonous bacteria (adherent bacteria) from six individuals of Northeast Arctic cod (NEAC) using high throughput shotgun DNA sequencing. We hope to provide helpful information that may contribute positively to, among others, the food industry. For example, in developing probiotics to reduce disease and mortality in aquaculture [26] or as a unique genetic resource in bioprospecting for, e.g., cold-active hydrolases.

## 2. Materials and Methods

### 2.1. Sampling of Fish and Intestinal Mucus

In January 2021, freshly collected intestinal tracts of Northeast Arctic cod were obtained from a fish factory outside of Tromsø, Norway (latitude 69.851620, longitude 18.821799). Intact gastrointestinal tracts (stomach-pyloric caeca-intestine-rectum-anus) were collected from the end of a processing line in the fish slaughterhouse during the processing of freshly landed catches of adult NEAC cod. The sex of the animals from which the organs originated is therefore not known. Figure 1A shows the intestinal tract of one specimen. The intestinal tracts were emptied of their lumen content (feces and transient bacteria) by gently squeezing the intestines before washing them with sterile saline solution three times. The intestines were then cut with a clean scissor and placed on a sterile dish. Using disposable plastic pipettes and spoons, any remaining feces were removed from the inside of ca. 10 cm section of the intestine segments (Figure 1B), and the mucus was collected in 2 mL Eppendorf tubes by scraping the inside surface of the intestines. The mucus samples were stored at −80 °C for up to six months before DNA extraction.

### 2.2. Isolation of DNA and Shotgun Sequencing

Total DNA was then isolated from the collected mucus sample of six Northeast Arctic cod. Three DNA isolation protocols (Figure 2) were tested and modified before DNA of sufficient quantity and quality was obtained.

Initially, total DNA was extracted using the DNazol method (Thermofisher Scientific, Paisley, UK) [27], followed by phenol:chloroform:isoamyl alcohol extraction and ethanol precipitation. This protocol was optimized with additional lysozyme treatment and RNase A digestion. However, despite satisfactory DNA yields and OD ratios, the Illumina sequencing using Swift Turbo library preparation was not successful. The presence of RNA in the samples, even after RNase A digestion, may have led to an overestimation of the DNA concentration. In a subsequent attempt, mucosal DNA was extracted using the High Pure PCR Template Preparation Kit (Roche, Basel, Switzerland). Unfortunately, this method failed to yield high-quality DNA and was not suitable for sequencing.

Finally, the FastDNA™ Spin Kit for Soil (MP Biomedicals, Oslo, Norway) was utilized to extract total DNA from the intestinal mucus of six fish, using 200–250 mg of starting material, following a method similar to that previously described [22]. Here, four tubes per fish were used to increase the DNA yield. At the end of the protocol, the samples were eluted with 200 µL of clean distilled water. In addition, RNA was removed using 1 µL of RNase cocktail (500 U/mL RNase A and 20,000 U/mL RNase T1 g/mL) (Invitrogen by Thermofisher Scientific, Oslo, Norway) for 10 min at 37 °C. At last, DNA purification began with phenol-chloroform extraction, adding 200 µL of phenol-chloroform to an equal volume of DNA solution and mixing by inversion. After centrifugation at room temperature for 5 min at 14,000 rpm, the lower phase was partially discarded to reduce phenol contamination and enhance DNA yield. A second centrifugation clarified the phase separation. The aqueous phase, about 180 µL, was transferred to a new tube, avoiding the phenol-chloroform phase. DNA was precipitated overnight with 1/10 volume of 3M sodium acetate and two volumes of cold 100% ethanol at −20 °C. After mixing by inversion and centrifuging at 14,000 rpm and 4 °C for 30 min, the supernatant was discarded. DNA pellets were washed with 500 μL of 70% ethanol, re-centrifuged, and the residual ethanol was removed after a quick spin. Pellets were dried using a Speed Vac and resuspended in 20 μL of 10 mM Tris buffer (pH 7.5). The purity of the samples was monitored by Nanodrop 2000c (Thermofisher Scientific, Waltham, MA, USA) and the final DNA concentration was determined with a Qubit 2.0 Fluorometer (ThermoFisher Scientific, Oslo, Norway). DNA Yields ranged from 0.29 to 10 ng/μL (average of 1.77 ng/μL), suitable for Illumina MiSeq sequencing, as shown in Table 1.

These samples were DNA sequenced at the Norwegian Sequencing Centre (NSC) using the Illumina MiSeq platform (https://www.sequencing.uio.no/illumina-services/, assessed on 27 October 2024). Due to the low yield of the isolated total DNAs, sequencing was done with 250 bp (Smart ThruPlex for low-input samples) paired-end reads. At first by subjecting samples to Smart ThruPlex library preparation for low-input samples, and then by running the 250 bp paired-end sequencing using MiSeq Reagent v.2 (500 cycles).

### 2.3. Quality Control of DNA Sequence Reads

Following DNA sequencing, we removed any reads that matched host (i.e., Atlantic cod) DNA or Illumina adapters using FastQ_Screen v.0.13.0 [28] and Trimmomatic v.0.39 [29], respectively. Furthermore, the quality of the sequence reads was accessed with FastQC v.0.11.9 [30], and low-quality reads were removed with Trimmomatic v.0.39. NGS statistics are provided in Table 2. Overall, the number of reads between the different samples varies by up to twofold, with an average of 13 million reads. The read lengths also varied from 54 to 118 bp (average of 92 bp). After quality control, 65 to 96.5% of the reads were identified as host DNA and removed. Furthermore, after trimming the remaining low-quality raw reads, an average of 1.9 million reads were kept per sample before further analysis.

### 2.4. Rarefaction Curves

Using the same technique as previously described, rarefaction curves were created for all six mucosal samples from the Northeast Arctic cod (see Figure A1) [22]. Using the Kaiju output files, which contained the identified bacterial families from the six fecal samples, rarefaction curves were created. Each output file was first transformed into “rarefaction reports”, which have many lines called reads (i.e., up to 3 million), where each line/read corresponds to a single bacterial family. As a result, one specific bacterial family can be represented by numerous lines or reads. Finally, yet importantly, rarefaction curves were created utilizing the “rarefaction report” from each distinct sample, where bacterial families were chosen randomly at various read counts ranging from 1000 to ca. 3 million reads, depending on the number of reads available for each distinct sample. A family, however, was only represented once and as a singular family in the rarefaction curves, even when it was found in a sample more than once.

### 2.5. Taxonomic Profiling and Metagenome-Assembled Genomes (MAGs)

After removing host DNA and quality trimming, we obtained an average of 1.98 million reads per sample with an average length of 92 bp. The processed sequencing reads were used for taxonomic classification using Kaiju version 1.6.2 [31] with default settings. We instructed Kaiju to analyze the reads using protein sequences from the MarDb and MarRef version 2 databases (specific databases for marine organisms) [32], as previously described [22]. Kaiju generated a detailed taxonomic report for the six samples, which included the names of bacterial families, and the corresponding number of reads assigned to each family. To determine the proportion of each bacterial family within the samples, percentages were calculated based on the number of reads classified, excluding any unclassified reads. This approach provided a clearer understanding of the relative abundance of different bacterial families in the samples, based solely on the reads that could be definitively identified. Next, we assembled the six metagenomes into longer contigs using MetaSpades v.3.15.3 [33] and attempted to produce MAGs using Maxbin v.2.2.7 [34]. All six samples were assembled successfully. However, after running Maxbin and adjusting the available options of the tool (e.g., contigs-length), we were able to obtain sufficient quality MAGs for only three samples (i.e., MBRG 49, MBRG 50, and MBRG 51). After trying to bin the assemblies with several various “length_contigs” options, we were able to optimize the bins for the three samples as follows: (1) 800 bp for MBRG-49, (2) 2000 bp for MBRG-50 and (3) 500 bp for MBRG-51. The obtained MAGs were further checked for completeness and contamination using CheckM v.1.0.12 [35] with default parameters. The criteria for validating and keeping bins were set to a completeness of at least 50% and contamination lower than 10%. Then, we used Sendsketch from the BBMap package v.38.84 [36] to identify the bacteria represented in the individual bins for each of the three samples. Sendsketch performs approximate taxonomic classification by using a hash function to create sketches (a sketch is a collection of k-mers, each typically 31-nt in size) that are compared to sketches made from reference genomes (Refseq). In order to check the identity results by sendsketch, we used BRIG v.0.95 (BLAST Ring Image Generator) with blast+ v.2.13.0 [37] that compared the bins against reference genomes using nucleotide fasta files. The Artemis Comparison Tool (ACT) v.18.0.2 by Carver et al. (2008) [38] was also used for comparing the MAGs to their corresponding references. First, the MAGs were reordered using abacas v.1.3.1 [39]. Then, the comparison file that is required for running the act was produced by using blastall v.2.9.0–2 on the fasta file from the MAG and its corresponding reference.

### 2.6. Phylogenomic Analyses of the MAGs

The EzTree pipeline v.0.1 [40] was used as previously described [22] for producing concatenated multiple sequence alignments from identified single-copy markers genes in reference genomes and MAGs. Finally, we used MEGA 11 [41] for creating the maximum likelihood (ML) phylogenetic trees from the sequence alignments as described earlier [22] and using the JTT+G+I evolutionary model, and the stability of nodes was tested with a bootstrap analysis (ML/JTT+G+I/200 pseudoreplicates).

## 3. Results

### 3.1. DNA Sequencing Revealed Three Different Taxonomic Profile Types Among Six Mucosal Samples from Migrating Northeast Arctic Cod

First, we tested the number of bacteria per unit of mass in the mucus of three fish and found that CFU/g ranged from 10^7^ to 10^8^. The results from the taxonomic profiling based on metagenomic sequencing are shown in Figure 3 and show that we found three very different taxonomic profiles among the samples. These were denoted “Type I”, “Type II”, and “Type III”.

Averaged over three samples (MBRG 46, MBRG 47, and MBRG 50), Type I is dominated in numbers by *Pseudomonaceae* (44%), followed by *Xanthomonadaceae* (13%). Averaged over two samples, Type II is highly dominated in numbers by one family, *Vibrionaceae* (65%). Finally, Type III comprises one sample only, which contains 76% representatives from *Enterobacteriaceae*. This was surprising to us since previous studies of >40 Atlantic cod consistently identified *Vibrionaceae* as the dominating bacterial family in the intestinal tract, either in the transient gut content (i.e., feces) [22,42] or in the mixed fecal/mucosal material [21,43,44]. Another notable observation is that, on average, 229 families were identified per sample. In addition, rarefaction curve calculations showed that for three of the six samples, the number of discovered bacterial families was saturated. For the remaining three samples, more sequence depth would likely result in a higher number of discovered families (Figure 1A). Both characteristics and ecological roles of the dominant bacterial families are summarized in Table 3 and further discussed in the discussion section.

### 3.2. Binning of Assembled Contigs Produce High-Completeness Bins/MAGs

To further study the most abundant bacteria in each mucosal sample, we next produced metagenome-assembled genomes (MAGs). We obtained five high-completeness bins/MAGs for three mucosal samples (see Table 4), where each sample represented one taxonomic profiling (MBRG 49 (Type II), MBRG 50 (Type I), and MBRG 51 (Type III)), as seen in Figure 3.

The remaining three mucosal samples only produced low-quality bins (due to limited read lengths and low number of classified reads) and were discarded (see Materials and Methods for more details). Additionally, we analyzed the five high-completeness MAGs/bins from samples MBRG 49 (Bins 1–3), MBRG 50 (Bin 1), and MBRG 51 (Bin 1) using Sendsketch from BBMap package [36]. Three high-completeness bins from sample MBRG-49 were all identified as *Photobacterium iliopiscarium*, a bin from sample MBRG-50 was identified as *Pseudomonas fluorescens*, and a bin from sample MBRG-51 was identified as *Shigella* sp./*Escherichia coli*, respectively. This result agrees with the most abundant families identified by Kaiju (see Figure 3), but further narrows the bacteria to genus or species level. The size of one bin per sample corresponds well with the expected sizes of reference genomes, i.e., 4.17 Mb (Bin 3 from sample MBRG-49) versus 4.26 Mb (*P. iliopiscarium* ATCC 51760), 7.7 Mb (Bin 1 from sample MBRG-50) versus 6.5 Mb (*P. fluorescens*), and 4.53 Mb (Bin 1 from sample MBRG 51) versus 4.55 Mb (*Shigella* PAMC 28760). Furthermore, BRIG analysis showed that the three MAGs were matching their respective references (Figure 4).

Moreover, we continued our genome-level comparison in more detail using the Artemis Comparison Tool [38]. The results are provided as a Appendix A. They show that MAG MBRG51_bin1 and the reference *Shigella* PAMC 28760 are highly similar, with only a few gaps between them. These gaps could represent indels (insertions or deletions) in either the MAG or reference, missing data in the MAG, or over-binning (contamination). The quality of MAGs MBRG49_bin3 (95% completeness and 3.2% contamination) and MBRG50_bin1 (96.5% completeness and 6.5% contamination) are overall lower and this is also reflected in the ACT comparison. We conclude that the current quality of the two MAGs is not sufficient to allow for detailed comparisons.

### 3.3. EzTree Robustly Places High-Completeness Bins on Maximum Likelihood Trees

The final protein datasets produced by EzTree consisted of 96, 282, and 81 concatenated marker genes, respectively. Figure 5 shows that MBRG-49_bin3 (Type II taxonomic profile) is found nested within the *Photobacterium phosphoreum* clade and is closely related to *P. iliopiscarium*. The placement of MBRG-49_bin3 (from the adherent microbiota) is identical to that of MAGs identified as part of the transient gut microbiota (see e.g., MBRG-38_bin1 and MBRG-30_bin1 in Figure 5). MBRG-50_bin1 (Type I taxonomic profile) is placed within the *Pseudomonas fluorescens* group and branches as a sister to *P. yamanorum* (a bacterium found in Antarctica) (Figure 5B). Finally, MBRG-51_bin1 (Type III taxonomic profile) is positioned within *Escherichia*/*Shigella*, and is most closely related to *Shigella* PAMC 28760 (Figure 5C).

## 4. Discussion

In this study, we describe for the first time the microbial composition of the mucosal tissues of six Northeast Arctic cod (i.e, “migrating Atlantic cod”) by using DNA extracted from intestinal mucus and performing next-generation shotgun sequencing (i.e., a metagenomics approach). We discovered three different taxonomic profiles, denoted as “Type I,” “Type II,” and “Type III,” among six specimens. These are dominated in numbers by *Pseudomonaceae* (44%), *Vibrionaceae* (65%), and *Enterobacteriaceae* (76%), respectively. Finally, we identified *P. fluorescens*, *P. iliopiscarium,* and *Shigella* as the most abundant bacterial species in the intestinal mucus of six Atlantic cod for each profile type. The general picture for mucosal samples is, therefore, different from what has been established in previous studies for the transient part (feces) of the intestine, where *Vibrionaceae* was consistently identified as the most abundant family, either in the transient gut content alone [22,42] or in the mixed fecal/mucosal material [21,43,44,66,67]. We conclude that even though we obtained limited sequence reads of bacterial origin, our data is sufficient for evaluating the most abundant families and the overall taxonomic profile of samples.

To gain a deeper insight into the characteristics and ecological roles of the most abundant bacterial families found in the mucus of cod intestines, we reviewed the literature. Firstly, *P. fluorescens* (abundant in Type I) is a ubiquitous bacterium found in soil, water, and plants, noted for its versatile metabolism and production of antimicrobial compounds [68]. In addition, the bacterium is found as part of the normal flora in the intestines of healthy fish [46,66,69], and here it has been observed to exert an antagonistic effect by offering protection against infections, and by contributing to the equilibrium of the gut ecosystem [45]. For example, a study by González-Palacios et al. (2019) revealed that two specific strains of *P. fluorescens* (strains LE89 and LE141), effectively decrease infections by the stramenophile *Saprolegnia parasitica*, a known pathogen of Rainbow trout [47]. Similarly, *P. fluorescens* has been reported to counteract *Flavobacterium psychrophilum*, which is responsible for high mortality rates in rainbow trout within aquaculture settings, as documented by Korkea-aho et al. (2012) [70]. Further supporting these findings, research by Eissa N et al. (2014) identified three biovars of *P. fluorescens* with antimicrobial properties against harmful pathogens, including *Pseudomonas anguilliseptica* and *Streptococcus faecium* [71]. These studies collectively suggest that *P. fluorescens* not only competes with other bacteria in the gut of Atlantic cod but can also play crucial roles in preserving the health of its host.

Secondly, *P. iliopiscarium* (abundant in Type II) can be part of the normal microbial community, particularly in marine animals [72]. It is usually monitored and studied because it is associated with seafood and meat spoiling [56,73,74]. The role of *P. iliopiscarium* in the fish gut is unfortunately not clear [44]. However, many members of *Photobacterium* genus (e.g., *P. phosphoreum*) are considered commensal or mutualistic, contributing to the host’s health or engaging in beneficial interactions [72,75]. For example, *P. phosphoreum* is commonly found in the gut of marine fish, where it is believed to play a role in the digestion of food by degrading chitin from crustacean prey [54,75]. In addition, several studies showed the antagonistic property of *Photobacterium* [44,55]. Our data support that the adherent microbiota from two fish includes *P. iliopioscarium* as the most abundant bacteria. The very presence of *P. iliopiscarium* in the mucosal layer of two fish in our study suggests that the bacterium may play a pivotal role in nutrient absorption and immune modulation, e.g., by synthesizing essential vitamins and facilitating the breakdown of complex dietary components, thereby directly influencing the host’s nutritional status and energy balance.

Finally, *Shigella* PAMC 28760 was found as the most abundant bacteria in the Type III profile. It should however be noted that the bioinformatic identification of *Shigella* is not definitive, with *E. coli* being an equally likely candidate. This is attributed to the close evolutionary and genetic relationship between *Shigella* and *E. coli* [76], the latter of which is typically non-pathogenic and a common constituent of the normal gut flora in warm blooded mammals [59,77]. *Shigella*/*E.coli* are also commonly identified in fish intestines [60,61]. *Shigella* PAMC 28760 was identified in Antarctic lichens [63].

Although the data is limited, our results show that the most abundant adherent bacteria vary between individual Atlantic cod, whereas previous data from us and others has firmly established that the transient microbiome is similar even between cod individuals sampled from various coastal locations at different time points and seasons [22,44]. We can only speculate why we observe conspicuous differences for the adherent bacteria between the six studies fish. Factors that contribute to the composition of fish microbiota has been suggested to include e.g., host selection, host genetics, developmental stage of the fish host, diet, and environment [78]. Host selection is used to explain why individuals of same species have similar microbiota despite being scattered into different environments and being exposed to different diets [78]. This explanation fits very well with the transient gut microbiome of Atlantic cod, which is highly dominated in numbers by *Photobacterium* strains (*Vibrionaceae*) regardless of sex, yearly season, and geographical location. However, the host selection hypothesis does not agree well with our current observations with highly variable adherent gut microbiomes.

Another possible major determinant for the gut microbiome is genetic diversity among Atlantic cod populations. Several studies have indeed revealed moderate to high genetic diversity between Atlantic cod populations that live in different zones of the same marine area [79,80]. For example, Kent and co-workers used 8076 SNPs to study genetic diversity among Atlantic cod populations in three different Baltic Sea regions [79]. They found a significant correlation between genetic diversity, and geographic distance and bottom salinity. The Barents Sea’s varying salinity is driven by riverine freshwater, the North Atlantic current’s saltier waters, and less saline Arctic inflows that create unique habitats [81,82]. To summarize, genetic differences among various groups of NEAC and environmental factors such as salinity and temperature might account for the observed variation in the adherent bacterial communities in their intestinal mucus. Finally, differences in diet among populations of cod due to the availability of different prey animals could also be a contributing factor [83]. For example, it is well documented that the composition of species varies spatially in the Barents Sea, many of which are on the diet of Atlantic cod [84,85].

How do our results from the mucosal microbiota of Atlantic cod compare to other studies? This is the first study to employ a metagenomic NGS approach to analyze the mucosal microbiota of Atlantic cod. While microbiota studies on the adherent bacteria from fish intestines already exist, such as in farmed rainbow trout from Scotland [86], farmed rainbow trout from Perthshire, UK [87], Arapaima gigas from eastern Amazon, Brazil [88], and grass carp from Hubei Province, China [26], these studies vary in focus and detail. For instance, in farmed rainbow trout from Scotland, *Enterobacteriaceae* was the prevalent group, constituting 20% of the total microbiota, with 86% of these being closely related to *E. coli*. Notably, *E. coli* was identified as a prevalent adherent bacterium, absent from the gut content, aligning with our findings. Other studies have reported microbial diversity at broader taxonomic levels, such as phylum or class, making direct comparisons with our detailed findings challenging. For example, amplicon sequencing of DNA isolated from the intestinal mucus of reared rainbow trout in the UK identified predominant bacteria, including Gammaproteobacteria (36.6%), Betaproteobacteria (18.4%), Bacilli (16.8%), Fusobacteria (10.8%), and Alphaproteobacteria (7.1%) [87]. In *Arapaima gigas* from the eastern Amazon, Brazil, prevalent bacteria such as *E. coli*, *Edwardsiella tarda*, *Citrobacter braaklii*, and *Pleisomonas shigelloides* have been reported [88]. Additionally, variations in the adherent gut microbiota of grass carp across different locations have been observed [26]. In summary, although the literature on bacterial diversity in intestinal mucus is sparse, our results partially align with those from relevant studies.

The rest of this discussion will focus on the technical challenges one may encounter when sequencing DNA that has been isolated from mucus samples, e.g., from the intestinal tract of fish. Using several different protocols, we were able to recover only small amounts of DNA from the outer mucosal layer of the cod intestine. Most of the DNA turned out to be from the host itself (65.5–96.5%; see Table 2) and not from colonizing bacteria. Similar proportions of host DNA have been reported by others when human mucosal intestinal samples were used [89]. Moreover, after removing sequence reads from host DNA and low-quality reads, we obtained relatively few and short sequences of bacterial origin (Table 2). Of these, 54% to 98% (an average of 72%) remained unclassified, which suggests either that similar sequences are not found in the current databases or that the sequences were too short to be robustly classified. Given these challenges and our difficulties in isolating a high concentration of metagenomic DNA, we need to consider possible contamination from exogenous DNA (i.e., extraction kits, reagents, tubes, etc.) that may skew the interpretation of our results. For example, DNA contamination during DNA extraction using commercial kits, especially for low biomass samples (i.e., blood, sputum, mucus), has been reported in microbiota studies [90,91]. Thus, we have further evaluated the likelihood of our mucosal samples being contaminated with bacterial DNA from the FastDNA^TM^ Spin Kit for soil (MP biomedicals), which was the DNA isolation kit used in this study. Firstly, we reviewed the literature on DNA isolation kit contamination risks, notably a study by Salter et al. (2014), who found significant contamination from *Burkholderiaceae* (about 40%) [92]. No *Vibrionaceae* contamination was detected, and *Enterobacteriaceae* comprised about 10% of reads, with *Pseudomonaceae* representing 20–30%. They also observed that diluting the sample reduced the number of target organism sequences with extremely low read counts (e.g., 210, 79, and less than 20 reads). In contrast, our study yielded significantly higher read counts, ranging from 208,834 to 2,852,610 total reads and 6720 to 1,603,839 classified reads. Secondly, we reassessed the most abundant MAGs by comparing them to the closest genomes in the databases, with the findings detailed below for each. For example, the most abundant and complete MAG (MBRG50_bin1) that belongs to *Pseudomonas fluorescens* family (Type I) exclusively represents marine bacteria and is relatively distantly related to non-marine *Pseudomonas*. The most abundant MAG in “Type II” (MBRG49_bin3) is part of the marine *Photobacterium phosphoreum* clade, previously identified in Atlantic cod by us and others. Finally, MBRG51-bin1 was found to be most closely related to *Shigella* PAMC 28760, which has previously been isolated from a cold marine environment, specifically from the *Himantormia* sp. lichen in Antarctica [63]. The BRIG and ACT genome-level comparisons further supported this close relationship (see Figure 4 and Appendix A). We therefore conclude that it is highly unlikely that MBRG51_bin1, which was identified as the marine *Shigella*, originates from a contamination from kits or other reagents. However, it should be noted that feces may not have been completely removed from the mucus samples, and therefore potentially influencing our results, e.g., *Vibrionaceae* was previously identified as the dominant family in the feces of Atlantic cod [21,22,44].

Despite the low number of classified reads, from a few samples (e.g., 1.8% classified reads for MBRG-47), we were able to recover enough reads from the DNA to establish the bacterial diversity and identify the most abundant bacteria in the intestinal mucus of Atlantic cod, as shown in Figure 3. Marine databases were used for identifying the bacteria present in our samples. Sequencing of 16S rDNA amplicons represents an alternative method to shotgun DNA sequencing, and it offers some advantages; for example, amplicon sequencing can be successfully done with much smaller amounts of isolated DNA, and the quality of the DNA is less critical since sequences are generated from DNA that has been PCR amplified and purified before sequencing. These are important benefits, and probably why it represents a widely used method. However, the downside is that only partial 16S rDNA sequences are generated, hence no information other than the taxonomic profile can be retrieved.

## 5. Conclusions

In conclusion, our study provides the first glimpses into the microbial diversity of the mucosal layer inside the intestinal tract of Northeast Arctic cod, using shotgun sequencing. However, technical challenges represented significant obstacles and limited the number of samples we were able to provide (with high confidence). Therefore, similar future studies would be of value to verify our results, which suggest that there are greater variations in the adherent microbial composition of the mucosal layer compared to that of the transient feces. In addition, future studies should include control measures such as the addition of a control DNA of known origin to determine the potential contamination of bacterial DNA from kits and a control with feces for assessing the potential impact of transient fecal bacteria. We also recommend collecting as much metadata as possible, such as sex, age, weight, and length of the fish.

Finally, the findings from this study offer potential applications across multiple fields, including ecology, ichthyology, and the food industry. In ecology, the identification of unique microbial profiles in the gut mucus of Northeast Arctic cod can assist our understanding of host–microbe interactions in marine environments and, in the long term, potentially contribute to the management of cod populations. In ichthyology, these results can provide background for future studies on fish gut microbiota; they can shed light on microbial roles in digestion, immunity, and adaptation to various environments. For the food industry, particularly in aquaculture, insights into the gut microbiota could lead to the development of probiotics aimed at improving fish health, reducing disease prevalence, and increasing the sustainability of fish farming.

## Figures and Tables

**Figure 1 microorganisms-12-02204-f001:**
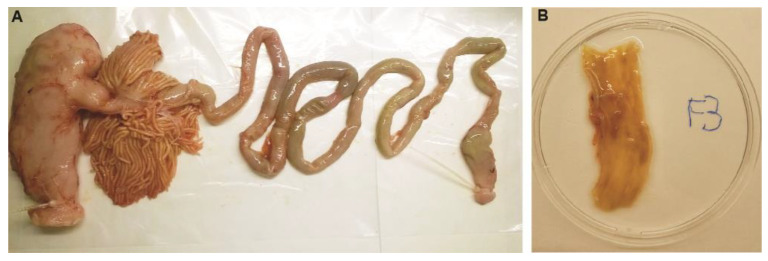
Photos showing the gastrointestinal tract of freshly obtained Northeast Arctic cod. (**A**) The intestinal tract was obtained from the end of the processing line at a fish factory near Tromsø, Norway (latitude 69.851620, longitude 18.821799). The stomach is to the left in the photo and was filled with mostly undigested capelins. The stomach is followed by the pylorus and pyloric caeca, then the intestine which ends in the rectum (or distal gut). (**B**) A section from the middle of the intestine was carefully opened and placed on a petri dish. The inner intestinal wall was rinsed and scraped with a clean plastic spoon to obtain the outer mucus layer.

**Figure 2 microorganisms-12-02204-f002:**
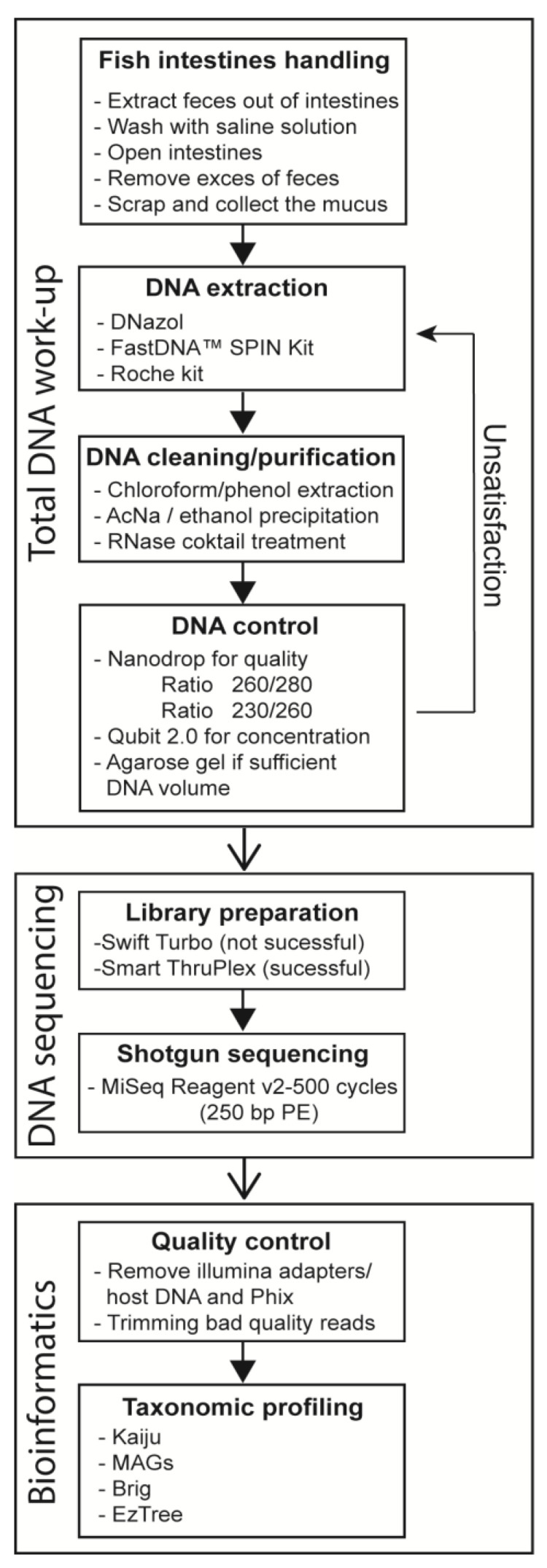
Schematic representation of the workflow from the handling of fish intestines to analyzing the sequencing data using bioinformatics.

**Figure 3 microorganisms-12-02204-f003:**
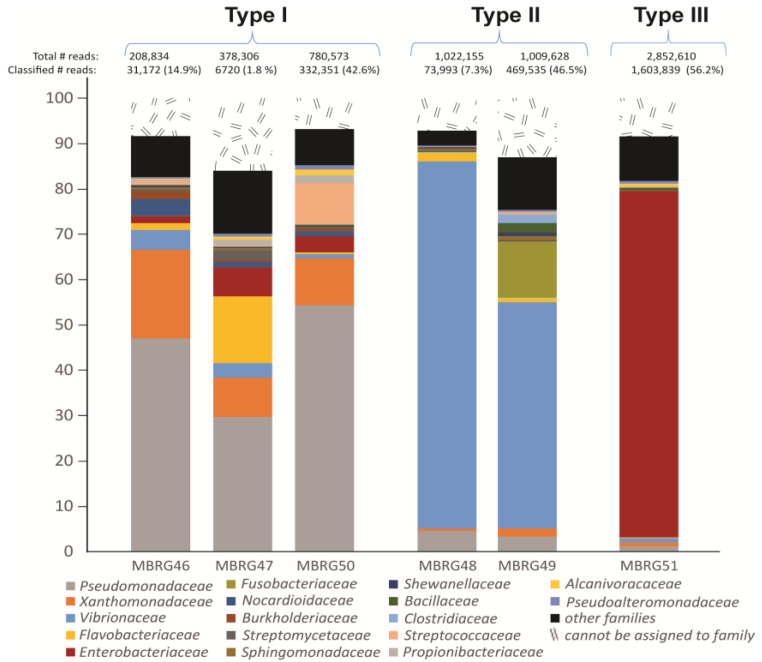
Bar chart showing the taxonomic profile of the adherent microbiota of the gastrointestinal tract of the Northeast Arctic cod. The most abundant families from six mucosal samples, i.e., MBRG-46 to MBRG-51 are shown. Data is based on shotgun DNA sequencing using an Illumina Miseq instrument and the V2 chemistry (250 bp end-pair reads). Taxonomic profiles were grouped into three “types”, i.e., Type I–Type III, based on their most abundant families.

**Figure 4 microorganisms-12-02204-f004:**
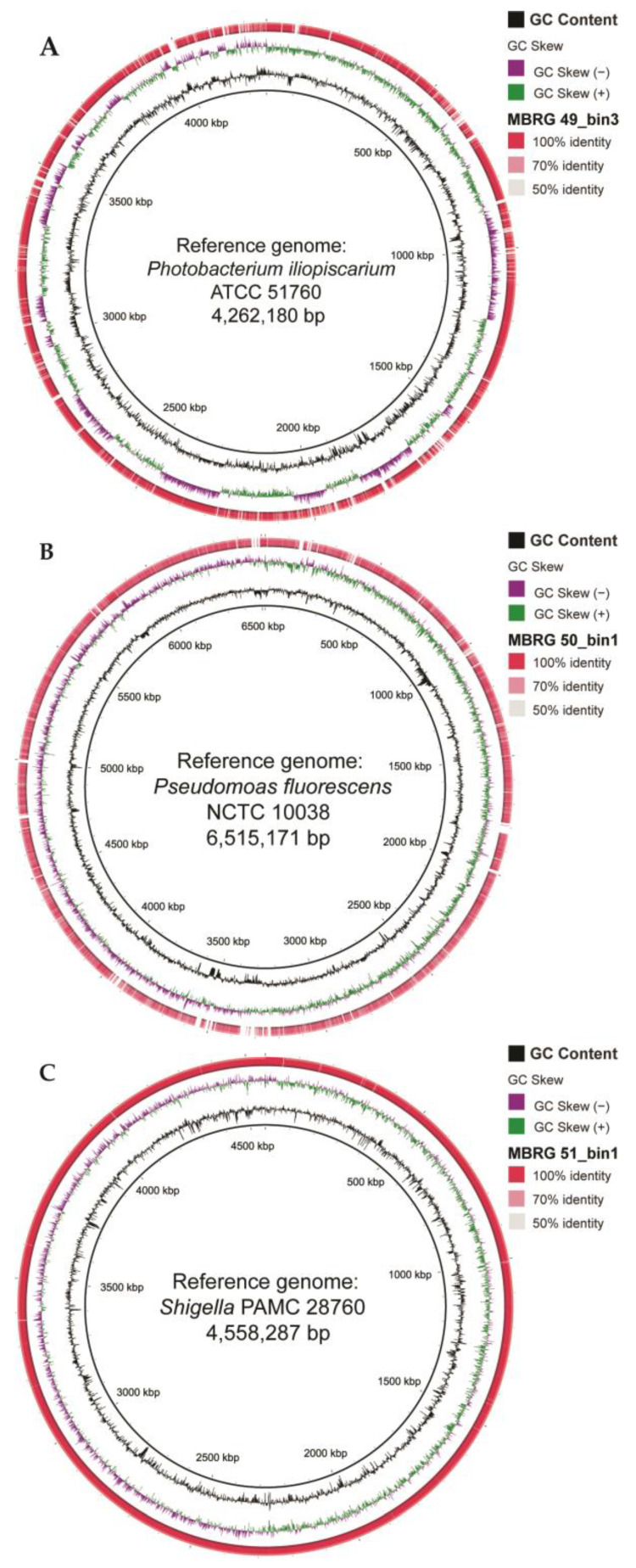
Comparison between MAGs from each taxonomic profile type and reference genomes. (**A**) MBRG-49_bin3 (from Type II) was blasted against the reference *P. iliopiscarium* ATCC 51760, (**B**) MBRG-50_bin1 (from Type I) was compared with the reference *P. fluorescens,* and (**C**) MBRG-51_bin1 (from Type III) was compared with the reference *Shigella* PAMC 28760. The figure was created with the BLAST Ring Image Generator (BRIG) tool. The center black ring represents the reference. GC content and GC skew are displayed as well. Colored rings symbolize the matches between MAGs and the reference, with color intensity showing the identity percentage (100%, 70%, or 50%).

**Figure 5 microorganisms-12-02204-f005:**
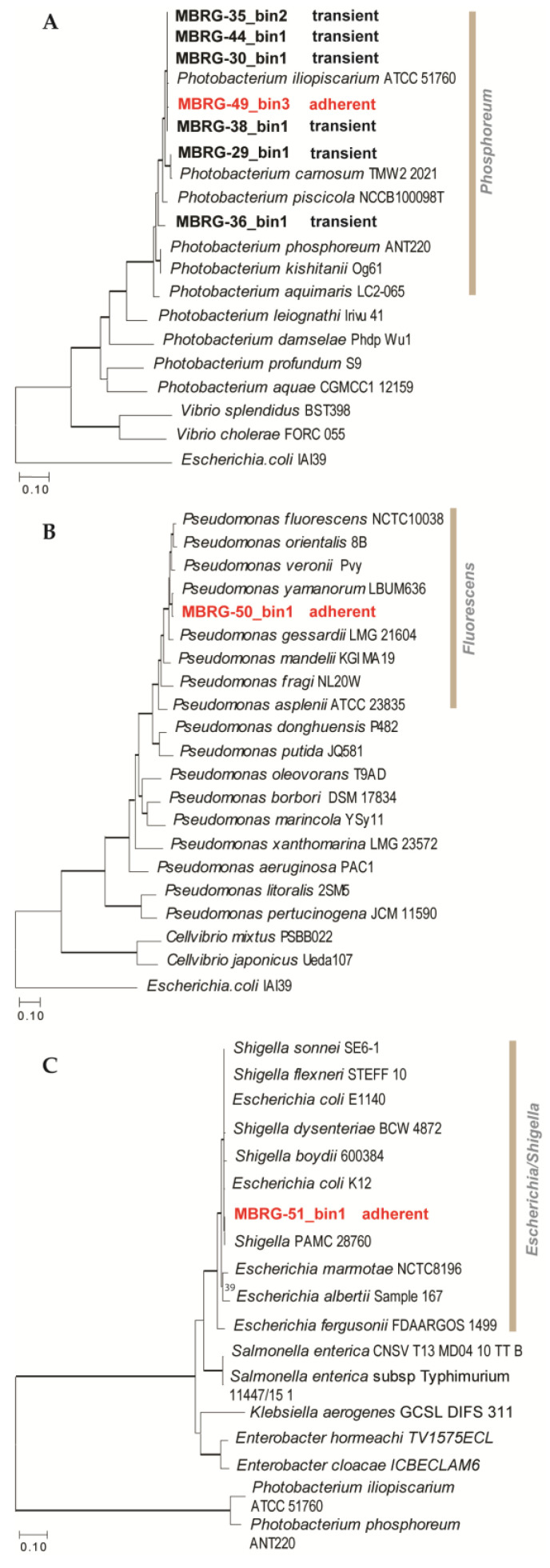
Phylogenomic analysis of MAGs and reference genomes from the databases. (**A**) The ML tree includes MBRG-49_bin3 (highlighted in red bold text), six MAGs from the transient part of the intestine of Atlantic cod [22], twelve reference *Photobacterium* genomes, and *E. coli* as the outgroup. (**B**) The ML tree shows the phylogenetic relationships between the MAG MBRG-50_bin1 and 19 reference genomes in addition to *E. coli* (outgroup). (**C**) ML tree of MBRG-51_bin1, 15 reference genomes, in addition to two outgroup genomes (*Photobacterium*). Thick branches represent strong bootstrap support (≥98%). MAGs from this study belong to clades highlighted with a grey vertical bar.

**Table 1 microorganisms-12-02204-t001:** Quality and concentration of DNA extracted from six intestinal mucus samples from Northeast Arctic cod.

Fish ID	DNA Quality		DNA Concentration in ng/µL ^1^		Available DNA for Sequencing	
	260/280	230/260	No RNase Treatment	Treated with RNase Cocktail ^2^	Volume (µL)	Concentration (ng)
MBRG46	1.8	1.83	2.56	0.29	15	4.35
MBRG47	2	1.6	2.09	0.45	20	9
MBRG48	1.86	1.84	17	10	19	19
MBRG49	1.95	1.72	19.4	0.70	10	7
MBRG50	1.91	1.73	1.6	0.11	12	1.34
MBRG51	1.89	1.4	0.261	0.14	19	2.6

^1^ The concentration of DNA was measured using Qubit 2.0. ^2^ RNase cocktail contains RNase A and RNase H.

**Table 2 microorganisms-12-02204-t002:** Sequencing information of DNA mucosal samples from the intestines of six Atlantic cod using the MiSeq Illumina platform.

Fish ID	Raw Data		Final Dataset	
	# Reads	Cod DNA (%)	# Reads ^1^	Average Length (bp)
MBRG46	9,805,968	96.5	327,852	116
MBRG47 *	12,534,644	95.4	541,404	54
MBRG48 *	21,894,396	94.7	1,496,529	57
MBRG49	8,376,018	76.4	1,964,746	106
MBRG50	10,955,154	86.2	1,491,889	118
MBRG51 *	16,305,536	65.5	5,543,182	101

* Asterisks denote the samples that were sequenced twice using Miseq Illumina. ^1^ Number of reads after removal of Cod DNA and trimming of the bad quality reads using trimmomatic v0.39.

**Table 3 microorganisms-12-02204-t003:** Summary of the characteristics and ecological roles of the most abundant bacterial families identified in the mucus of six Atlantic cod.

Family Name	Characteristics	Ecological Roles	Species in MAGs *	References
*Pseudomonadaceae*	➢ Commensal/Pathogenic ➢ Heterotrophic bacteria ➢ Predominantly aerobic ➢ Ubiquitous bacteria: in soil, water, and on plants. Also found in human and animals ➢ Versatile metabolism ➢ Antibiotic-producing bacteria	➢ Antagonistic activity ➢ Protects against infections and diseases by producing ATB and bacteriocin ➢ Contributes to gut ecosystem equilibrium ➢ Promote plant growth ➢ Bioremediation	*Pseudomonas fluorescens*	[45,46,47,48,49,50,51,52]
*Vibrionaceae*	➢ Commensal/Pathogenic ➢ Predominantly of facultative anaerobes ➢ Fermentative bacteria ➢ Natural inhabitant of marine environments and associated with marine animals	➢ Antagonistic activity towards pathogens in fish ➢ Symbiotic properties ➢ Role in nutrient cycling ➢ Aiding in food digestion (i.e., chitin degradation from crustacean preys) ➢ Host’s nutritional status and energy balance. ➢ Meat spoiler	*Photobacterium iliopiscarium*	[22,44,53,54,55,56,57,58]
*Enterobacteriaceae*	➢ Commensal/Opportunistic pathogen ➢ Predominantly of facultative anaerobes ➢ Inhabiting the gastrointestinal tract of human and animals, soil, vegetation, and marine environments	➢ Antagonistic activity ➢ Regulate microbial communities in complex microbial ecosystems (i.e., microcin production in human’s intestine) ➢ Preserve a stable anaerobic condition in animal guts ➢ Vitamins production	*Shigella* PAMC 28760	[59,60,61,62,63,64,65]

* The MAGs (Metagenome assembled genomes) and their predicted bacterial origin are present below (Section 3.2).

**Table 4 microorganisms-12-02204-t004:** Representation of the complete MAGs for each profiling type identified in the intestinal mucus of the Northeast Arctic cod.

	MAXBIN						CHECKM		SENDSKETCH		
Sample ID	Bin ID	Rel Abund ^1^(%)	Contigs (n)	Comp ^2^ (%)	Genome Size(bp)	GC Content (%)	Comp ^2^ (%)	Cont ^3^ (%)	Bacteria IDs	KWID (%)	KID (%)
MBRG49	1	52.5	88	96.3	1,796,209	28.5	100	0	*Photobacterium iliopiscarium*	1.2	0.3
	2	34.7	274	90.7	1,902,394	30.7	98.6	0	*Photobacterium iliopiscarium*	12.7	3.6
	3	12.8	1396	98.1	4,168,186	41.7	95	3.2	*Photobacterium iliopiscarium*	80.8	50.4
MBRG50	1	36.4	653	99.1	7,662,027	60.8	96.5	6.5	*Pseudomonas fluorescens*	78.1	59.2
MBRG51	1	85.3	126	99.1	4,535,891	50.6	99.4	0.2	*Shigella* sp.	99.9	49.8

Information for each obtained MAG includes relative abundance of bins, contig number, genome size, GC content, completeness and contamination from Maxbin and CheckM, bacterial ID from sendsketch (RefSeq database’s closest matches), % KID (kmer match between query and reference), and KWID (normalized % KID to genome size). ^1^ The abundance of bins as calculated by Maxbin. ^2^ Genome completeness from Maxbin and CheckM. ^3^ Contamination scores from CheckM.

## Data Availability

The metagenomics datasets generated during the current study are available under the BioProject number PRJEB59225 and secondary accession number ERP144281 in the European Nucleotide Archive (ENA) [93] at https://www.ebi.ac.uk/ena/browser/view/PRJEB59225 (accessed on 2 September 2023). The six metagenomics raw sequences have been deposited as BioSample from SAMEA112445163 to SAMEA112445168.

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
