# Peer review of "The Microbiota of the Outer Gut Mucus Layer of the Migrating Northeast Arctic Cod (Gadus morhua) as Determined by Shotgun DNA Sequencing"

_microorganisms, 2024, doi:10.3390/microorganisms12112204_

Round 1

Reviewer 1 Report

Comments and Suggestions for Authors

Use of shotgun metagenomics is a powerful way of assessing microbial community structure and functioning. In the current paper the authors make use of this technique to investigate autochthonous bacteria present in the cod gastrointestinal tract. In the study three different taxonomic profiles, denoted “Type I” “Type II” and “Type III”, were revealed for analyses of among six cod specimens. The authors argue the microbial community structure in the mucosal samples is different from what has been established in previous studies for the transient part (feces) of the intestine, where Vibrionaceae was consistently identified as the most abundant family. If this is correct this is an imporant finding which contributes to our understanding of the cod microbiome. 

The authors conclude that 'provides the first glimpses into the microbial diversity of the mucosal layer inside the intestinal tract of Northeast Arctic cod'. This might be true, but it should be noted that a more comprehensive analyses could have been completed with the avialable data. For example, it would be interesting to see how the genetic composition of the detected MGAs differ from their closest relatives as this might provide some clues about adaptation to the gastrointestinal tract.

A major concern is the low number of bacterial reads obtained. The authores explain their difficulties with optaining enough DNA for sequencing. Moreover, among sequenced reads, up to 96% percent  were identied as fish-derived. With low bacterial DNA content, one is vulnerable to contamination of bacteria from for example extraction kits. Indeed, several of the taxonmic groups identified in the metagenomes have been deteced in DNA extraction kits (see for example https://gutpathogens.biomedcentral.com/articles/10.1186/s13099-016-0103-7/tables/2), such as Enterobacteriaceaev(Escherichia/Shigella), Pseudomonadaceae, Xanthomonadaceae, Flavobacteriaceae, Burkholderiaceae, Sphingobacteriaceae, Bacillaceae, Clostridiaceae, Streptococcaceae, Propionibacteriaceae, Alcanivoracaceae.

The potential for contamination should be discussed / clarified and authors should explicitely discuss why some of the dominating MAGs should not be regarded as contaminants from kits etc. 

Despite the efforts to remove transient (feces) from the sample material, I would presume that it is difficult to know if this was 100% successful. The potential for 'contamination' of transient bacteria should also be discussed in more detail.

Without further analyses of the contamination issues it is difficult to evaluate the significance and soundness of the presented results. In the worst case, (e.g.  if it is highly uncertain if the reads obtained are mostly contaminants from kits), the paper should not be published as it might create more confusion than new insight. 

It could have been advantageous for the paper to include 
- a control of known DNA (e.g. only fish DNA run through the same extraction pipeline as the test samples) to evaluate the potential input of bacterial DNA from kits etc.
- a fecal samples control to evaluate the potential input of fecal DNA (transient bacteria).

Minor comment:

Line 212: escribed >> described

Author Response

Reviewer 1

#1

The authors conclude that 'provides the first glimpses into the microbial diversity of the mucosal layer inside the intestinal tract of Northeast Arctic cod'. This might be true, but it should be noted that a more comprehensive analyses could have been completed with the avialable data. For example, it would be interesting to see how the genetic composition of the detected MGAs differ from their closest relatives as this might provide some clues about adaptation to the gastrointestinal tract.

We did the following to compare the MAGs with reference genomes in further detail: (i) arranged the MAG contigs according to reference genomes using ABACAS (ref.), (ii) then subjected the reordered MAG and corresponding ref. genome to Blastall (ref.) to create a comparison file, and (iii) used the three files (MAG, ref. genome and comparison file) as input to ACT (Artemis Comparison Tool). The results are provided as supplementary Figure (Fig. S1) and show that MAG MBRG51_bin1 and the reference Shigella PAMC28760 (see also comment #2) are highly similar with only few gaps between them. These gaps could represent indels (insertions or deletions) in either the MAG or reference, or simply missing data in the MAG, or over-binning (contamination). But, overall, the comparison reflects that MAG MBRG51_bin1 is of high quality (99.4% completeness and 0.2% contamination), and highly similar to Shigella PAMC 28760.

The quality of MAGs MBRG49_bin3 (95% completeness and 3.2% contamination) and MBRG50_bin1 (96.5% completeness and 6.5% contamination) are overall lower and this is also reflected in the ACT comparison. Of the raw cumulative size of the bins, approx. 0.78 Mbp and 3.43 Mbp, respectively, did not match well to the corresponding references. Although it is an appealing idea, the current quality of the two MAGs will probably not allow for detailed studies on how the bacteria from which the two MAGs originate, are adapted to the gastrointestinal tract. At least it would be outside the scope of this study.

The ACT comparison is described in Fig. S1 and Lines 306-315 in the manuscript.

#2

A major concern is the low number of bacterial reads obtained. The authores explain their difficulties with optaining enough DNA for sequencing. Moreover, among sequenced reads, up to 96% percent  were identied as fish-derived. With low bacterial DNA content, one is vulnerable to contamination of bacteria from for example extraction kits. Indeed, several of the taxonmic groups identified in the metagenomes have been deteced in DNA extraction kits (see for example https://gutpathogens.biomedcentral.com/articles/10.1186/s13099-016-0103-7/tables/2), such as Enterobacteriaceaev(Escherichia/Shigella), Pseudomonadaceae, Xanthomonadaceae, Flavobacteriaceae, Burkholderiaceae, Sphingobacteriaceae, Bacillaceae, Clostridiaceae, Streptococcaceae, Propionibacteriaceae, Alcanivoracaceae.

The potential for contamination should be discussed / clarified and authors should explicitely discuss why some of the dominating MAGs should not be regarded as contaminants from kits etc.” 

Thank you for this very valuable comment, and potential contamination from kits, reagents, or even from the feces etc. should indeed be discussed. Your feedback has heightened our awareness of contamination risks in studies involving samples with low biomass.

We have addressed the concern about potential contamination as follows:

First, we re-evaluated the most abundant MAGs with respect to their closest genomes in the databases, the findings are described below for each of them.

Type 1 – The most abundant and complete MAG (MBRG50_bin1) groups within the Pseudomonas fluorescens family, which represents marine bacteria only. Moreover, the MAG is relatively distantly related to non-marine Pseudomonas.

Type 2 – The most abundant and complete MAG (MBRG49_bin3) groups within the Photobacterium phosphoreum clade, which represent marine bacteria, and has been identified in Atlantic cod by us and others in the past.

Type 3 - The most abundant and complete MAG (MBRG51_bin1) groups among Escherichia/Shigella. In the Phylogenomic tree (Fig. 5C) this MAG groups as sister to Shigella PAMC 28760. Next, we added two marine E. coli genomes and calculated the pairwise distances between all possible pairs of sequences using the JTT-model (see picture below). The smallest distance (0.000001) in the analysis is between MAG MBRG51_bin1 and the Shigella PAMC 28760. Shigella PAMC 28760 was isolated from a marine environment (from Himantormia sp. lichen in Antarctica). In conclusion, MBRG51_bin1 is closely related to a marine Shigella from a cold environment and it is therefore highly unlikely that it originates from contamination from kits or reagents.

The ACT comparison, in the new Supplementary file Fig. S1, also support the close relationship between MBRG51_bin1 and Shigella PAMC28760 (also image below).

Furthermore, we did a literature study on the risks of contaminations from DNA isolation kits etc. Salter et al 2014 (BMC Biol. 12:87) studied contaminants from DNA isolation kits including the kit we used in this study (Fast DNA spin kit for soil, MP Biomedical) and found that it contained predominantly with Burkholderiaceae (approximately 40% in three sample dilutions). Vibrionaceae was not detected, and Enterobacteriaceae was found only in about 10% of total reads, excluding the reads for the bacteria supposed to be present in their study. Pseudomonaceae accounted for 20% to 30%. They demonstrated that the more diluted the test sample was (with Salmonella bongori), the fewer sequences are assigned to S. bongori, and more are assigned to others. However, the number of reads sequenced was extreme low (e.g., 210, 79, and less than 20 reads). In our study we obtained 208,834 to 2,852,610 number of reads in total, and 6720 to 1,603,839 classified reads, which is considerably higher than that reported in Salter et al. 2014.

In conclusion, the MAGs from our study from the most abundant reads are highly similar to marine/aquatic bacteria, which strongly suggest to us that they do not originate from contaminations from kits. We acknowledge that we cannot be entirely certain that all feces were removed from the mucosal sample. This has been added to the discussion (lines 440-480).

#3

“Despite the efforts to remove transient (feces) from the sample material, I would presume that it is difficult to know if this was 100% successful. The potential for 'contamination' of transient bacteria should also be discussed in more detail”.

Agreed. See response #2 above.

#4

“It could have been advantageous for the paper to include 
- a control of known DNA (e.g. only fish DNA run through the same extraction pipeline as the test samples) to evaluate the potential input of bacterial DNA from kits etc.
- a fecal samples control to evaluate the potential input of fecal DNA (transient bacteria)”.

Thank you for this suggestion. Unfortunately, we are unable to redo these experiments and add a control of known DNA and a fecas samples to control. We have added this as a recommendation to future studies. (see Lines 500-504).

Minor comment:

#5

“Line 212: escribed >> described”

We have added more description as suggested (Material and Method:”Taxonomic profiling and Metagenome-assembled genomes (MAGs)”.

See Lines 204 to 210

Reviewer 2 Report

Comments and Suggestions for Authors

Good manuscript! 

The three types of microbial profiles (Type I, Type II, and Type III) are clearly defined, but the implications of this variation are not fully explored. Are there environmental factors, diet, or migration patterns that might explain these differences? A more thorough discussion of these profiles would strengthen the narrative.

The figures are generally well-designed and informative. However, some could benefit from larger labels and clearer legends. For instance, in Figure 3, the distinction between the three taxonomic profiles could be emphasized more, perhaps with different colors or patterns.

Table 1 (DNA isolation outputs) and Table 3 (MAG representation) are well-organized, but an additional table summarizing the characteristics and ecological roles of the dominant bacterial families would be useful.

Author Response

Reviewer 2:

#6

The three types of microbial profiles (Type I, Type II, and Type III) are clearly defined, but the implications of this variation are not fully explored. Are there environmental factors, diet, or migration patterns that might explain these differences? A more thorough discussion of these profiles would strengthen the narrative”.

Thank you for your comment. In the discussion, we have already speculated on the possible reasons for variations in the most abundant bacteria in the intestinal mucus of NEAC among the six individuals studied. Realizing that our initial discussion was unclear, we have rephrased this part for better clarity.

See Lines 391 to 404

#7

“The figures are generally well-designed and informative. However, some could benefit from larger labels and clearer legends. For instance, in Figure 3, the distinction between the three taxonomic profiles could be emphasized more, perhaps with different colors or patterns

Thank you for your feedback. We have modified Figure 3 and improve so it is easier to directly see the difference between the three taxonomic profile “types”.

#8

Table 1 (DNA isolation outputs) and Table 3 (MAG representation) are well-organized, but an additional table summarizing the characteristics and ecological roles of the dominant bacterial families would be useful”.

The suggested table has been added to the manuscript as Table 3. It summarizes our literature study on both characteristics and ecological of the dominant bacterial families identified in the mucus of Atlantic cod. See lines 265-268

Reviewer 3 Report

Comments and Suggestions for Authors

The manuscript (Microorganisms-3250715) under the title "The microbiota of the outer gut mucus layer of the migrating Northeast Arctic cod (Gadus morhua) as determined by shotgun DNA sequencing" investigated the microbial composition of the mucosal tissues in Northeast Arctic cod by DNA shotgun sequencing. Results from this study confirmed there were three types of taxonomic profiles in the adherent gut microbiome of Atlantic cod. The predominance bacteria in the Type I, Type II, and Type III is Pseudomonaceae (44%), Vibrionaceae (65%) and Enterobacteriaceae (76%), respectively.

The main content of this paper is valuable for microbial diversity (adherent and non-adherent bacteria) in the intestinal tract of wild and farmed fish. However, the current paper contains too many redundant statements and unnecessary descriptions in the text, particularly the result part. There are several mistakes in language, syntax, and format. Additionally, it is suggested to use the latest Microorganisms template file to prepare the revised manuscript. Therefore, the authors should re-check and reorganize the whole manuscript to improve its quality before re-submission.

Major comments:

1. In the "1.Introduction" part, multiple descriptions were irrelevant and redundant. For example, the statements in Line 67-71 are correct, but they have nothing to do with the core theme of this study. They could be deleted without influencing the meaning of the corresponding paragraph. Furthermore, some sentences with confusing syntactic structure are confusing or hard to understand, such as Line 51-53. The authors should re-check "1.Introduction" part and rephrase the relevant statements for laying out more clearly background in this study.

2. In the "2.Materials and Methods" section, many methodological descriptions are too complicated and detailed. For example, the description on DNA extraction, shotgun sequencing, taxonomic profiling and MAGs, particularly in Line 121-165. The description in Line 115-116 is correct, but a bit superfluous. Moreover, some sentences in the text of "2.Materials and Methods" are a bit colloquial. It is recommended to shorten the corresponding part in "2.Materials & Methods" or move the relevant descriptions to the supplementary part.

3. There were too many unnecessary and redundant descriptions in "3.Results". As an example, in Line 245-247, Line 263-265, etc, the original texts are speculative and discursive statements. It would be preferable to move these sentences to the discussion part. There are many same errors in the other parts of "3.Results".

Several statements in this part are the methodological descriptions. For example, the text of Line 240-242, Line 272-275, etc. They could be deleted directly without any negative influence on the corresponding paragraph or moved to the "2.Materials and Methods" part.

Thus, the main content of the result part should be streamlined and written with emphasis. Please modify accordingly.

4. The current text of "4.Discussion" part was not well-organized. The discussion could be more in depth in terms of the similarities and differences of microbial composition between your study and relevant research in the intestinal tract of Northeast Arctic cod or other fish species. Some sentences in the original text of "3.Results" can be moved and merged into the existing text of discussion part.

Please reorganize and rephrase the section of "4.Discussion" to emphasize and better clarify the major findings in this study.

5. Please check the references format carefully according to the instructions for authors. Multiple mistakes are present in the current reference list, with several page numbers and volume numbers missing or wrong, along with other inconsistencies like capitalized vs. lower-case article titles, italicized vs. non-italic latin name of species.

For example, the page numbers and volume numbers should be revised or provided in Reference 32, 33, 36, etc. In Reference 9, 15, etc., the presentation of page number is inconsistent.

Reference 21, 35, and 38 have non-italic scientific name of species in article title. There were similar errors in the other references.

In Reference 67, it is expressed as the capital letters for all words of article title.

Additionally, there are only 24 cited literatures (total literatures: 68) published in 2019-2024. Please make sure about 50% of the references are within 5 years (2019-2024).

The authors should re-check and modify the reference list seriously.

Minor comments:

1. Please check the symbols for volume unit in this study according to the information to related guides. For example, in Line 145, please replace "μl" with "μL". There were same mistakes in the other part of the manuscript. The authors need to check and revise accordingly.

2. What are the body weight and body length of six Northeast Arctic cod used for this study? The authors should give the relevant information in the "2.Materials and Methods" section.

Other errors (highlighted in yellow) were marked in the PDF file.

So, this manuscript will be reconsidered after major revision.

Author Response

Reviewer 3:

 Major comments:

#9

“In the "1.Introduction" part, multiple descriptions were irrelevant and redundant. For example, the statements in Line 67-71 are correct, but they have nothing to do with the core theme of this study. They could be deleted without influencing the meaning of the corresponding paragraph”.

Done. We have removed the statements as suggested.

#10

“Furthermore, some sentences with confusing syntactic structure are confusing or hard to understand, such as Line 51-53. The authors should re-check "1.Introduction" part and rephrase the relevant statements for laying out more clearly background in this study”.

Fixed. Statements were reformulated for clearer explanation as suggested.

#11

“In the "2.Materials and Methods" section, many methodological descriptions are too complicated and detailed. For example, the description on DNA extraction, shotgun sequencing, taxonomic profiling and MAGs, particularly in Line 121-165. The description in Line 115-116 is correct, but a bit superfluous. Moreover, some sentences in the text of "2.Materials and Methods" are a bit colloquial. It is recommended to shorten the corresponding part in "2.Materials & Methods" or move the relevant descriptions to the supplementary part”.

Thank you for your comment. We have shortened the section “ 2.2 Isolation of DNA and shotgun sequencing” in the Materials and Methods as suggested. We have kept the original text and added it to the supplementary file.

#12

“There were too many unnecessary and redundant descriptions in "3.Results". As an example, in Line 245-247, Line 263-265, etc, the original texts are speculative and discursive statements. It would be preferable to move these sentences to the discussion part. There are many same errors in the other parts of "3.Results".

Several statements in this part are the methodological descriptions. For example, the text of Line 240-242, Line 272-275, etc. They could be deleted directly without any negative influence on the corresponding paragraph or moved to the "2.Materials and Methods" part.

Thus, the main content of the result part should be streamlined and written with emphasis. Please modify accordingly”.

Thank you for your comment. We have removed sentences that were too speculative and discursive from the “Results”. Lines 248-249, 236-265 and 266-267 were deleted.

Lines 245-247 moved to discussion.

We also removed sentences that were describing the methods, and some of these sentences were added to Materials and Methods. For example, we deleted from “Results”: lines 240-241, line 278, lines 288-293, 311-313, 315-322 and 323-325.

Lines 290-293 were moved to “Materials and Methods”.

We also removed parts of the sentences in lines 272-275. However, we kept “to further study the most abundant bacteria in each mucosal sample, we next produce metagenome assembled genomes (MAGs)” because it provides a better flow and understanding of the results to the reader.

We kept lines 301-302 with modification so the sentence describes the result obtained from BRIG. And we removed lines 311 to 313.

We have checked the pdf file from you and fixed the comments accordingly.

#13

“The current text of "4.Discussion" part was not well-organized. The discussion could be more in depth in terms of the similarities and differences of microbial composition between your study and relevant research in the intestinal tract of Northeast Arctic cod or other fish species. Some sentences in the original text of "3.Results" can be moved and merged into the existing text of discussion part.

Please reorganize and rephrase the section of "4.Discussion" to emphasize and better clarify the major findings in this study.”

Agreed. We have reorganized the discussion as advised, now starting with discussing the main findings, including a paragraph comparing our result with a few other studies (Lines 420-439), then moved on to discussing the technical challenges. Finally, we added recommendations and direction for the future.

#14

“Please check the references format carefully according to the instructions for authors. Multiple mistakes are present in the current reference list, with several page numbers and volume numbers missing or wrong, along with other inconsistencies like capitalized vs. lower-case article titles, italicized vs. non-italic latin name of species.

For example, the page numbers and volume numbers should be revised or provided in Reference 32, 33, 36, etc. In Reference 9, 15, etc., the presentation of page number is inconsistent.

Reference 21, 35, and 38 have non-italic scientific name of species in article title. There were similar errors in the other references.

In Reference 67, it is expressed as the capital letters for all words of article title.

Additionally, there are only 24 cited literatures (total literatures: 68) published in 2019-2024. Please make sure about 50% of the references are within 5 years (2019-2024).

The authors should re-check and modify the reference list seriously”.

Thank you for your feedback on the reference list. We have downloaded the «MDPI.ens references style file” from the EndNote website and used it. Now the volume numbers and page numbers should be correct.

We have also added more recent references to the paper. We also kept older ones because they are also relevant and additional to the new ones.

Minor comments:

#15

“Please check the symbols for volume unit in this study according to the information to related guides. For example, in Line 145, please replace "μl" with "μL". There were same mistakes in the other part of the manuscript. The authors need to check and revise accordingly”.

Fixed

#16

“What are the body weight and body length of six Northeast Arctic cod used for this study? The authors should give the relevant information in the "2.Materials and Methods" section”.

We agree that the body weight and body length of the studies NEAC will be relevant information but unfortunately, we do not have this information. This is because the fish intestines were obtained “from the end of a processing line in the fish slaughterhouse, during processing of freshly landed catches of adult NEAC cod.

#17

“Other errors (highlighted in yellow) were marked in the PDF file”.

Fixed

Reviewer 4 Report

Comments and Suggestions for Authors

The paper microorganisms-3250715 presents data for the bacterial profiles of autochthonous bacteria from the migrating Atlantic cod using a shotgun DNA sequencing approach. The paper is well written, organized and adds new understanding to the the microbiota of the outer gut mucus layer of marine fish consumed by residents in the North.

Abstract - It is expedient to add a brief explanation of the relevance of the choice of the outer gut mucus layer as the main object of research.

Introduction - Written quite well, however in my opinion information could have been added about cod and its role in the study of microorganisms associated with fish. How unique is this fish species in this context. And what does the applied metagenomic approach bring to the development of the food industry globally?

Methods - This section is very comprehensive, including a description of a modified technique to extract DNA of the required quality. However, at the beginning of the first section (2.1) it is not stated how much time elapsed between the capture of the fish and the storage of the sample at -80 oC? Could this have influenced the results obtained? In addition, the characterisation of the fish (age/weight/sex/presence of parasites) would also add value to the paper. Given the limited number of samples, it would probably also be valuable to determine the number of bacteria per unit mass of each sample.

The discussion and conclusion lack at least prospectively practical recommendations for the use of the fundamental results obtained in ecology, ichthyology or the food industry.

I do not have any major concerns, and recommend consideration for inclusion in MICROORGANISMS.

Author Response

Reviewer 4:

The paper microorganisms-3250715 presents data for the bacterial profiles of autochthonous bacteria from the migrating Atlantic cod using a shotgun DNA sequencing approach. The paper is well written, organized and adds new understanding to the the microbiota of the outer gut mucus layer of marine fish consumed by residents in the North.

#18

“Abstract - It is expedient to add a brief explanation of the relevance of the choice of the outer gut mucus layer as the main object of research”.

Fixed. We have added information on why we have focused on the outer gut mucus layer, and slightly modified the rest of the text to not exceed 200 words as required by MDPI microorganisms.

#19

“Introduction - Written quite well, however in my opinion information could have been added about cod and its role in the study of microorganisms associated with fish. How unique is this fish species in this context. And what does the applied metagenomic approach bring to the development of the food industry globally?”

We have added some information in the introduction why studying the gut microbiota of Atlantic cod using metagenomics can provide interesting information for the food industry (for example in aquaculture).

See Lines 95-98

#20

“Methods - This section is very comprehensive, including a description of a modified technique to extract DNA of the required quality. However, at the beginning of the first section (2.1) it is not stated how much time elapsed between the capture of the fish and the storage of the sample at -80 oC? Could this have influenced the results obtained? In addition, the characterisation of the fish (age/weight/sex/presence of parasites) would also add value to the paper. “

The samples were frozen for six months because we spent a long time optimizing the DNA extraction protocols and also sent the samples twice for sequencing in Oslo. The freezing conditions of the samples (frozen versus unfrozen) may influence which bacteria are more easily extracted due to differences in cell wall structure between Gram-positive and Gram-negative bacteria (Agata WA et al 2014). The efficiency of extraction also depends on the method used, whether mechanical or chemical disruption during the lysis steps. Previous studies have shown that bead beating is more effective on fresh samples than on frozen ones for lysing Gram-positive bacteria. In our case, we also added lysozyme during the lysis step to aid in disrupting the cell walls of these bacteria, which might not be optimally lysed by following only the kit instructions.

We added in “M&M” that we did have the samples frozen up to 6 months (Line 113)

See also response to #16.

#21

“Given the limited number of samples, it would probably also be valuable to determine the number of bacteria per unit mass of each sample”.

Thank you for this comment. That is a good advice. After looking at the data, we remembered that we did calculate the number of CFU/g for the Mucus samples of at least 3 fish (MBRG47, 48 and 51). We used for that Marine agar plates to select for marine bacteria. We got respectively, 30, 20 and 100 million CFU/g. Unfortunately, it is not available for the three other samples.

We added CFU/g for 3 fish in lines 241-242

#22

“The discussion and conclusion lack at least prospectively practical recommendations for the use of the fundamental results obtained in ecology, ichthyology or the food industry”.

Thank you. A brief comment on this was added in the conclusion. See lines 505 to514)

Round 2

Reviewer 1 Report

Comments and Suggestions for Authors

The authors have done a great job in clarifying the potential ecological role of the detected groups and have discusses the potential of contamination in an adequate way. I see no reason to not accept the paper in its current form. 

Reviewer 3 Report

Comments and Suggestions for Authors

The resubmitted paper (microorganisms-3250715) entitled "The microbiota of the outer gut mucus layer of the migrating Northeast Arctic cod (Gadus morhua) as determined by shotgun DNA sequencing" has been revised as suggested. The authors reply to the comments of the reviewers one by one and explain how they revised the manuscript (in cover letter).

Regarding the unchanged contents in the revised version, the authors have given the corresponding explanation in the list of response.

But there are some minor errors in the reference list. As an example, in Reference 22, 23, 24, etc, the scientific name of fish species in article title should be italicized. The authors should check and revise the list of cited literature once more.

Although the current revision is suitable for acceptance, it still needs text editing to avoid minor errors.

In my view, the current manuscript is suitable for acceptance although it still needs text editing to avoid minor errors.